# A Comprehensive Analysis of In-Line Inspection Tools and Technologies for Steel Oil and Gas Pipelines

**Berke Ogulcan Parlak \*** and **Huseyin Ayhan Yavasoglu**

Department of Mechatronics Engineering, Yildiz Technical University, Istanbul 34349, Turkey
\* Correspondence: boparlak@yildiz.edu.tr

**Abstract:** The transportation of oil and gas through pipelines is an integral aspect of the global energy infrastructure. It is crucial to ensure the safety and integrity of these pipelines, and one way to do so is by utilizing an inspection tool called a smart pig. This paper reviews various smart pigs used in steel oil and gas pipelines and classifies them according to pipeline structure, anomaly-detection capability, working principles, and application areas. The advantages and limitations of each sensor technology that can be used with the smart pig for in-line inspection (ILI) are discussed. In this context, ultrasonic testing (UT), electromagnetic acoustic transducer (EMAT), eddy current (EC), magnetic flux leakage (MFL), and mechanical contact (MC) sensors are investigated. This paper also provides a comprehensive analysis of the development chronology of these sensors in the literature. Additionally, combinations of relevant sensor technologies are compared for their accuracy in sizing anomaly depth, length, and width. In addition to their importance in maintaining the safety and reliability of pipelines, the use of ILI can also have environmental benefits. This study aims to further our understanding of the relationship between ILI and the environment.

**Keywords:** in-line inspection; smart pig; sensor; oil; natural gas; pipeline; environment

## 1. Introduction

Today's main energy sources are petroleum and natural gas. If the primary energy consumption in the United States of America (USA) is analyzed based on sources, oil accounts for 35% of the total and natural gas accounts for 23% [1]. The transportation of these energy sources is important in terms of accessing energy. Pipelines are considered the most efficient way to transport oil and gas resources [2]. Furthermore, it surpasses other modes of transportation such as road, train, air, and sea when traveling long distances in terms of convenience, cost, safety, and environmental friendliness. As a result, almost all natural gas is transported via pipelines. For instance, about 97% of the natural gas and oil transported in Canada is carried through pipelines. As of 2017, the entire length of gas and oil pipelines in the world is estimated to be approximately 3,550,000 km, with gas pipelines measuring 2,965,600 km in length and oil pipelines measuring 584,000 km [3].

Oil and natural gas pipelines spanning kilometers are susceptible to accidents for a variety of reasons. The main causes of pipeline accidents are corrosion, gouges, plain and kinked dents, smooth dents on welds, smooth dents with other types of anomalies, manufacturing defects in the pipe body, girth and seam weld defects, and cracking [4]. If these anomalies are not addressed, they may result in pipeline failures such as leaks or ruptures, resulting in increases in costs, environmental risks, and even catastrophic accidents. According to the data of the US Department of Transportation Pipeline and Hazardous Materials Safety Administration (PHMSA), there were 681 accidents labeled as serious in the USA between 2002 and 2021. In these accidents, 260 people lost their lives, 1112 people were injured and USD11,043,742,158 financial losses occurred. Apart from that, as a result of the rupture of the pipeline in Michigan in July 2010, 1,000,000 barrels of oil leaked and caused great damage to the environment [5].

Pipeline integrity (PI) programs must be used to maintain the gas supply and prevent accidents. PI is threatened by many factors. These factors can be listed as mechanical [6], operational [7], natural [8], and third-party [9]. Periodic assessment of PI is needed to prevent pipeline accidents and thus higher costs, environmental risks, and fatal accidents. Methods used for the assessment of PI are hydrostatic pressure testing [10], direct assessment [11,12], and in-line inspection (ILI) [13].

Hydrostatic pressure testing is used to locate leaks and confirm the performance and durability of pipes, tubing, and coils [14]. During testing, the pipeline is typically filled with water, and the pressure is maintained above the maximum operating pressure for a period of time. Meanwhile, critical anomalies in the pipeline cause leakage. This proves that anomalies that do not leak in the pipeline are not critical and that it is safe to operate the pipeline at maximum operating pressure. Testing can damage the pipeline, especially when performed at levels higher than 100 percent of the specified minimum yield strength of the pipe material, so ILI is often preferred over hydrostatic testing [15].

Direct assessment is the inspection of pipelines by operators. Operators should combine their knowledge of the pipeline section's physical properties and operating records with the results of the examination, inspection, and evaluation [16]. Direct assessment is effective on limited anomalies such as internal corrosion, external corrosion, and stress corrosion cracking (SCC). Therefore, ILI offers a more comprehensive assessment than direct assessment.

The only inspection technology that offers extensive information regarding anomalies that do not pose an immediate threat to the PI is ILI [17]. The ILI tools can classify anomaly types and specify their orientation, size, and location [18]. There are numerous ILI robots that specialize in detecting metal loss, cracks, geometry deformations, leaks, and wax deposition [19]. These robots were classified as pig type [20], wheel type [21–23], caterpillar type [24,25], wall-press type [26], walking type [27], inchworm type [28], and screw type [29] by Choi and Roh [30].

Pigs are the most-preferred robots in the ILI industry due to their features such as longer operation time, less downtime, being driven by product flow, and compatibility with developing sensor technologies. Pigs are essentially cylindrical electronic tools equipped with a traction system that completely covers the inner wall of the pipeline, a battery system that provides energy for the pig, and sensors that detect anomalies in the pipeline. These electronic tools can be used as geometry tools, mapping tools, metal-loss tools, and crack-detection tools [31]. Furthermore, smart pigs can be classified according to many factors. Figure 1 summarizes these factors.

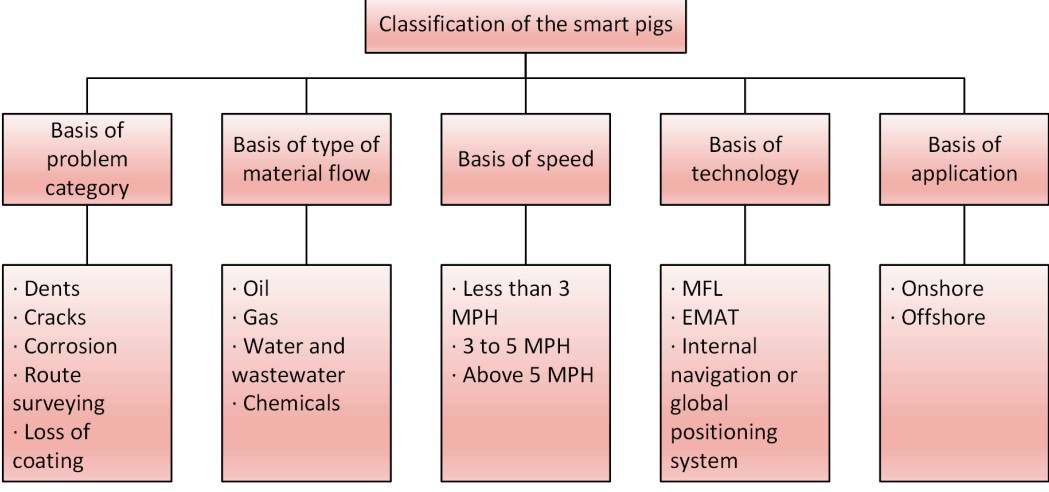

**Figure 1.** Classification of smart pigs [32].

Pipelines are classified as piggable or un-piggable. Pig operations can be conducted without difficulty in pipelines that are piggable. In un-piggable pipelines, the pig cannot be operated due to factors limiting the mechanical characteristics of the pig such as sharp turns, consecutive bends, and tee transitions in the pipeline. In addition, piggable lines may become un-piggable over time for various reasons [33]. These reasons can be listed as debris accumulation, configuration changes, and corrosion formation in the pipeline. Technologies such as closed-circuit television cameras (CCTV) [34] and smart ball [35] are used in un-piggable pipelines [36]. For piggable pipelines, the smart pigs that have been summarized in the literature by Song et al. [37] are the most suitable tool for the ILI of pipelines because of their speed, the sensor technology they can carry, and their ability to survey very long distances in one go. In the existing literature, the smart pig types and their sensor technologies have not been comprehensively evaluated in terms of anomaly-detection capabilities, historical developments, and hybrid sensor topologies.

This study aims to address the lack of understanding regarding the causal relationship between pipeline anomalies, pig-adaptive sensor technologies, and the environment in the current literature. The core of this study is based on a conference paper [38] presented at the 3rd Latin American Conference on the Sustainable Development of Energy, Water, and Environmental Systems, which includes literature reviews on basic ILI sensors. The main contributions of this paper are as follows:

- First, the ILI methodology was evaluated to current state-of-the-art developments.
- Second, a comprehensive comparison of non-destructive testing sensors specific to smart pigs was also conducted, taking into account their current technological status.
- Third, the ability of sensor technologies to detect various anomaly types and the use of hybrid sensors were also discussed in this paper.
- Finally, the relationship between the ILI and environmental impacts conducted within the scope of pipeline integrity management (PIM) was examined.

Following this introductory information, Section 2 provides the causes, hazard potentials, and management methods of pipeline anomalies. Section 3 includes the current status of ILI sensor technologies, as well as the advantages and limitations of basic pig-adaptive sensors, their working principles, and development chronologies in the literature. The anomaly-detection capabilities of ILI sensors and their hybrid structures are extensively evaluated in Section 4. The relationship of ILI sensors to the environment is given in Section 5 and finally, Section 6 presents the conclusion.

## 2. Types of Pipeline Anomalies

Anomalies occur in oil and natural gas pipelines for various reasons. The formation process of these anomalies can be examined in three classes. While corrosion anomalies in the pipeline occur depending on time, anomalies caused by mechanical damage or disasters occur independent of time. Anomalies originating from fabrication such as bends, buckles, and wrinkles are classified as stable. The management process of anomalies is associated with these formation processes. Therefore, the formation process of anomalies is important for PIM.

PIM is a systematic approach to ensuring the safe operation of pipelines. This process involves identifying and mitigating potential hazards that could lead to pipeline accidents. PIM can be divided into three main categories: assessment, planning, and management. Assessment involves close observation of the internal and external sections of pipelines to identify any anomalies and determine the overall condition of the pipelines. Planning encompasses all the activities aimed at maintaining or repairing pipelines, such as defining operations and procedures, conducting inspections, and performing maintenance and monitoring. Management includes tasks such as data management audits, fit-for-service evaluations, burst pressure assessments, and third-party verification. Through this comprehensive approach, PIM plays a vital role in protecting the environment and communities.

Pipeline anomalies can cause structural stress on the pipeline and increase the risk of pipeline failure. As such, it is important to examine the causes of occurrence, hazard

potentials, and management methods of pipeline anomalies as part of PIM. Walker [39] grouped the most important pipeline anomalies as geometric deformation, metal loss, and cracking. This paper provides comprehensive information regarding the formation and management process of these anomaly groups.

### 2.1. Geometric Deformation

The pipeline may be subject to operational distortions such as pressure fluctuations, excessive mechanical forces, poor workmanship, or third-party damage. These distortions are the main source of geometric deformations in the pipeline. Geometric deformations in the pipeline can be listed as metal movement, denting, metal removal, cold working of the underlying metal, and puncturing [40]. Among them, the dent is one of the three most common typical anomalies (i.e., corrosion, dents, and cracks) encountered in oil and gas pipelines [41]. Dents are formed on the surface of pipes due to external loads such as excavation activities during the construction of pipelines [42]. Dents can be classified as plaint dents [43] and composite dents [44]. Plaint dents pose no major threat to PI, whereas composite dents pose a greater threat to PI [45].

Dents cause stress and strain concentration [46,47]. Therefore, dent management is an important issue for PIM. Different approaches can be used for dent management. Warman et al. [48] presented an approach that Duke Energy Gas Transmission has implemented for dent management. This approach involved characterizing dents and mechanical damage in the pipeline system by integrating data collected from high-resolution ILI tools operated over 2000 miles. Torres and Piazza [49] developed a new engineering tool for the integrity management of dents using finite element analysis (FEA). This tool was used to develop fatigue life trends. There are other studies on the management and characterization of geometric deformations in the literature [50–52]. However, the most common use for detecting geometric deformations is pigs with mechanical contact (MC) probes.

### 2.2. Metal Loss

The pipeline may be exposed to rusting, cavitation, and corrosive substances over time. These reasons are the main source of metal losses in the pipeline. Metal loss can manifest itself as gouging, corrosion, and erosion. Among them, corrosion is one of the most important problems affecting safety in oil and gas pipelines, as shown in Figure 2, and accounts for approximately 30% of all equipment failures [53]. Corrosion continually reduces the pipe-wall thickness and can significantly accelerate the formation of leaks [54].

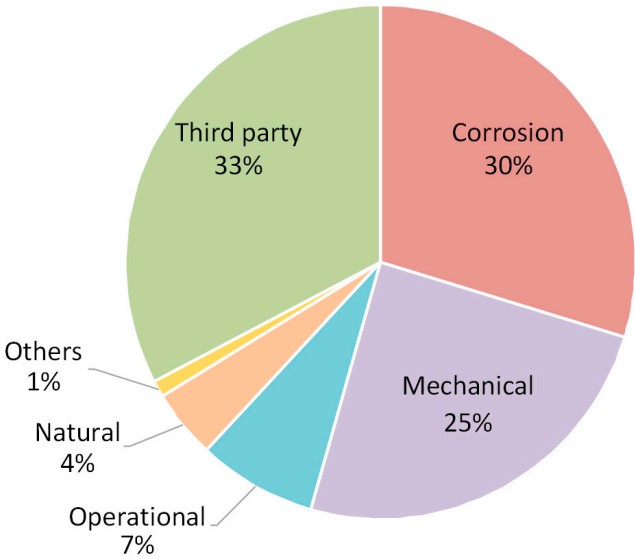

**Figure 2.** Percentages of oil pipeline failure types [55].



Accurate estimation of the corrosion rate is of primary importance for integrity management planning of a pipeline [56]. Machine-learning-based approaches are suitable approaches to predict the behavior of corrosion occurring in pipelines [57]. Hamed et al. [58] proposed a nonparametric calibration model based on k-nearest neighbor interpolation to improve field data collected from ultrasonic testing (UT) and magnetic flux leakage (MFL). The model improved the accuracy of pipeline wall-thickness measurements. In this way, the life of the critical part of the pipeline can be better predicted.

ILI tools are widely used for corrosion detection in oil and gas pipelines [59]. ILI is accepted as the optimum approach to detect and characterize the anomalies as well as reveal the growth rate information of the active anomalies in the pipeline [60]. Low and Selman [61] outlined the capabilities and limitations of ILI methods used to inspect corroded pipelines. Huyse et al. [62] presented a study that tested the performance of various ILI methods on top of the line (TOL) corrosion. According to the study, MFL technology showed the best success in detecting TOL corrosion. Palmer and Schneider [63] revealed that hybrid sensor technology, which combines different ILI methods such as UT, electromagnetic acoustic transducer (EMAT), MFL, and eddy current (EC), will be effective in sizing complex metal losses.

### 2.3. Cracking

Pipelines are constantly exposed to environmental effects, external loads, and ground movements due to their nature. These effects are the main source of cracking in pipelines. In addition, cracks often occur in a hybrid form in oil and natural gas pipelines. Examples of these hybrid anomalies are crack in corrosion (CIC), SCC, and crack in dent (CID).

Crack-like anomalies may occur simultaneously with corrosion anomalies and represent a new hybrid form of an anomaly called CIC [64]. Bedairi et al. [65] presented a study for predicting the failure pressures of CIC anomalies to determine the applicability of PI assessment methods. The predicted failure pressures were conservative when compared to the experimental results, with a mean difference of 17.4% for five different CIC anomalies with various depths.

SCC is defined as the growth of crack formation in a corrosive environment. These cracks often have a high aspect ratio and pose a major threat to the PI [66]. As a result, estimating the crack growth rate (CGR) of SCCs is critical. Song [67] developed a mathematical model for this purpose. The developed model was used to predict CGRs with two methods called the potentiodynamic polarization curve and the Butler–Volmer equation. The potentiodynamic polarization curve was good at predicting high CGRs, while the Butler–Volmer equation was good at predicting low CGRs. Ryakhovskikh and Bogdanov [68] determined the conditions for operating a pipeline with SCC cracks by considering the temporal variations of the CGR to certain accident statistics. The findings showed that pipes with crack depths between 0.1 and 0.25 $\delta$ (where $\delta$ is the pipe-wall thickness) could be left operating until a scheduled inspection if the CGR is estimated. Palmer et al. [69] presented a case study on CGR estimation based on repeated EMAT data. The method simply involved passing the EMAT sensor over the relevant crack periodically. Estimated CGRs determined during the presented case study showed reasonable results in line with the available literature.

Dents adjacent to welds can cause cracks to develop in the welds and cause a combined anomaly referred to as a CID or dent–crack anomaly [70]. These anomalies can lead to major accidents such as bursts in the pipeline. The effect of the location of the CID anomaly on burst pressure has been discussed in the literature [71–73].

ILI is frequently used in the management of oil and gas pipeline cracks. UT has proven to be the most suitable and reliable technology for crack detection in pipelines [74]. However, the application of UT technology requires a liquid medium. Therefore, EMAT is used as a substitute for UT technology in gas pipelines.

### 3. In-Line Inspection Sensor Technologies

The ILI system collects data from internal pipelines and is a key component of the pipeline industry's integrity management system that promotes safe, efficient, and cost-effective pipeline operations [75]. ILI technologies are constantly evolving depending on developments in sensor technologies. INGU Solutions has developed an inspection ball called Pipers [76], which includes a built-in three-axis accelerometer, gyroscope, magnetometer, and pressure and temperature sensors. The working principle of Pipers is similar to that of MFL, but the inspection ball uses the earth's magnetic field instead of the magnetic field created by a permanent magnet. While the inspection ball is low-cost and easy to operate, its measurement accuracy is low compared to conventional pig-adapted ILI sensors.

Pig-adapted novel ILI sensors are frequently encountered in the literature. Sampath et al. [77] introduced a new pig-adapted non-contact optical sensor array method for real-time inspection and non-destructive evaluation (NDE) of pipelines. The proposed sensor array included simple light-emitting diodes to send light to the pipeline's inner wall and light-dependent resistors to receive reflected light. The new array was successful on deposits, cavities, and uniform corrosions. Feng et al. [78] developed a novel alternating current field measurement probe that can be integrated into pigs and used in the inspection of natural gas and oil pipelines. The proposed probe consisted of sensors, supports, and inductive components (core and coils) and showed good results in corrosion and cracks. Sampath et al. [79] developed a smart pig that detects metal loss and geometric deformations using an optical sensor and bimorph sensor arrays. The proposed bimorph sensor array is based on the piezoelectric principle and consists of a bimorph sensor, probe tip, and cantilever beam components. When the probe tip passes over a geometric deformation, the cantilever beam is bent, the bending strain is measured by the bimorph sensor, and anomaly characterization is performed. The results showed that the proposed ILI method could accurately identify the anomaly size and location.

Despite extensive research on novel pig-adapted ILI sensors, these technologies tend to be inferior in practical application compared to the longstanding and preferred basic ILI sensor technologies in the industry. Basic ILI sensor technologies can be classified as UT [80], EMAT [81], EC [82], MFL [83], and MC [84]. This paper presents the methodology, limitations, and recent developments of basic ILI sensor technologies.

### 3.1. Ultrasonic Testing

UT is typically performed using a handheld probe that is passed over the surface of the material being inspected. The probe emits sound waves that travel through the material and are reflected back to the probe. The time it takes for the sound waves to travel through the material and be reflected back to the probe is measured, and this information is used to detect anomalies in the material. Figure 3 illustrates the functioning principle of a UT sensor.

In comparison to other technologies, ultrasonic is currently the most reliable ILI technique [85]. It is more feature-sensitive than MFL and gives better results on thick-walled pipes. UT waves can detect discontinuities in materials such as metals and polymers both above and below the surface. However, it is only used in liquid media due to limitations in its methodology. In addition, the pig should not accelerate to high speeds due to the difficulty of coupling the sensor to the pipe wall. The anomalies that UT can detect are internal and external metal losses, flanges, cracking, welds, etc.

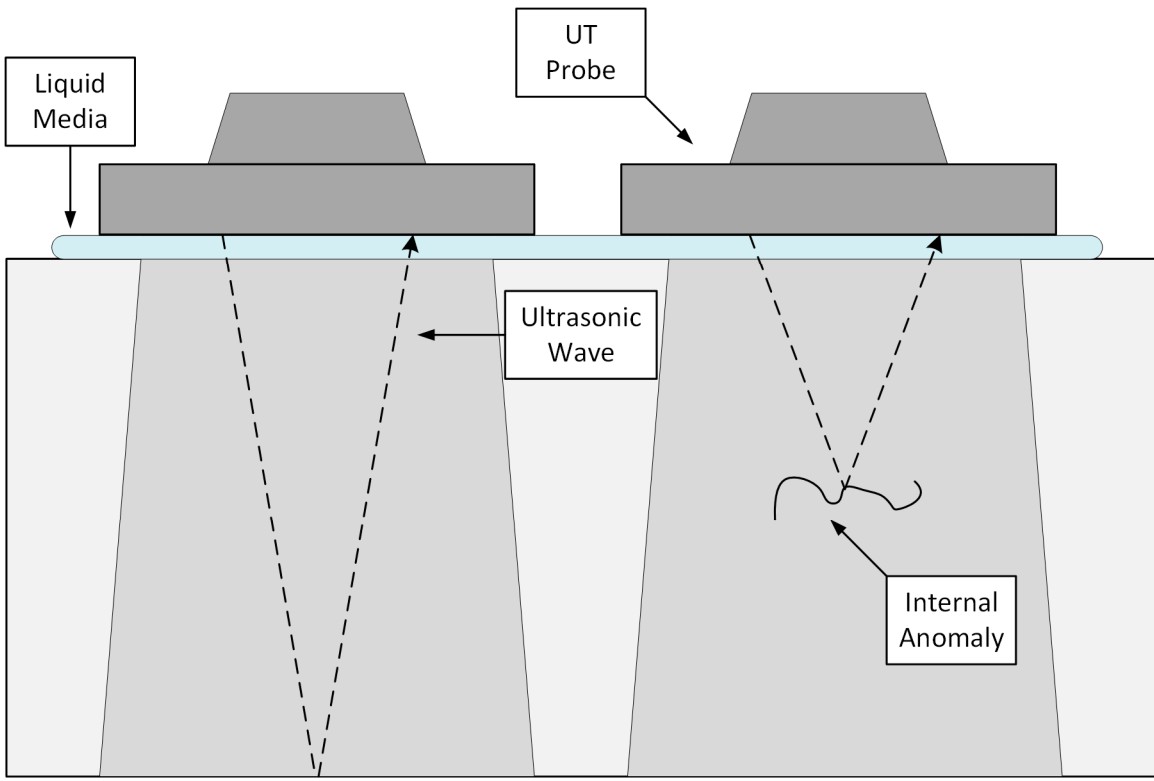

**Figure 3.** The working principle of a UT sensor.

The development of UT technology on different anomalies has occurred over time. The first ILI tool using ultrasonic technology for crack detection was introduced in the early 1990s [86]. This tool allowed inline detection of both internal and external cracks. Reber et al. [87] presented a design that made it possible to use the same tool for metal-loss or crack inspection. In this design, the same basic instrument assembly and electronic module could be used for both configurations by changing the sensor carrier. Significant savings in mobilization and demobilization costs have been achieved as a single tool can be used for both inspection tasks. However, two physically separate operations were still necessary. Reber et al. [88] later introduced another design that eliminated this problem and offered both metal-loss and crack detection in one operation. Thanks to this new array, higher inspection speeds, improved resolution, and accuracy have been achieved compared to the previous design. Dobmann et al. [89] examined the performance of UT at pipe welding points. Ultrasonic sensors arrayed in different structures according to axial, spiral, and girth weld types successfully detected welds in an investigation mission to detect transverse anomalies in an offshore pipeline. The internal rotary inspection system (IRIS), an ultrasonic pulse/echo immersion technique, first introduced by MatEval [90] in the early 1980s, was used by Birchall et al. [91] for anomaly detection in pigs. As a result of this study, fundamental pipeline anomalies such as external erosion, internal dent, and internal corrosion have been successfully detected. Slaughter et al. [92] presented a case study on the new-generation UT ILI crack tool called Combo WM CD. This tool showed improvements in high sensitivity, reduction in signal losses, higher resolution, probability of detection (POD), and anomaly sizing. UT technology can also be used to detect anomalies in unreachable locations in the pipeline. There are ultrasonic techniques such as higher-order mode cluster [93], multi-skip [94], and S0 mode lamb wave [95] in the literature. Khalili and Cawley [96] used these techniques to detect corrosion at unreachable points in the pipeline and classified them according to their success. Of these techniques, the higher-order mode cluster was very little affected due to its low surface motion and showed the best overall performance.

UT tools generally have three resolution components. These are axial resolution, circumferential resolution, and depth resolution. Willems [97] examined the developments in UT technology developed in the second decade of the 21st century in terms of hardware, data digitization, data processing, and data storage, and attributed the increase in these three basic resolutions to these concepts. Of these concepts, data storage has always been a problem because the UT tool contains too many transducer systems [98]. In this case, reducing the ultrasound data may be the solution. In the literature, there are techniques such as entropy coding [99], transformation techniques [100], techniques based on behavioral information of ultrasound signals [101], and FPGA-based architecture techniques [102]. The most efficient of these methods is the FPGA-based architecture technique, with an average data reduction of 96.5%.

### 3.2. Electromagnetic Acoustic Transducer

EMAT is a UT method that generates ultrasonic waves in the material being inspected rather than using a transducer. The EMAT sensor employs an electromagnetic field to produce ultrasonic waves. This field is generated by an electrical current that flows through a coil. The electromagnetic field excites the surface of the material being tested, causing it to oscillate. As the surface oscillates, it creates ultrasonic waves that travel along the surface of the material. The receiver circuit in the EMAT sensor detects these ultrasonic waves and thus the anomaly is characterized using the frequency, amplitude, and other properties of the wave. The working principle of an EMAT sensor is given in Figure 4.

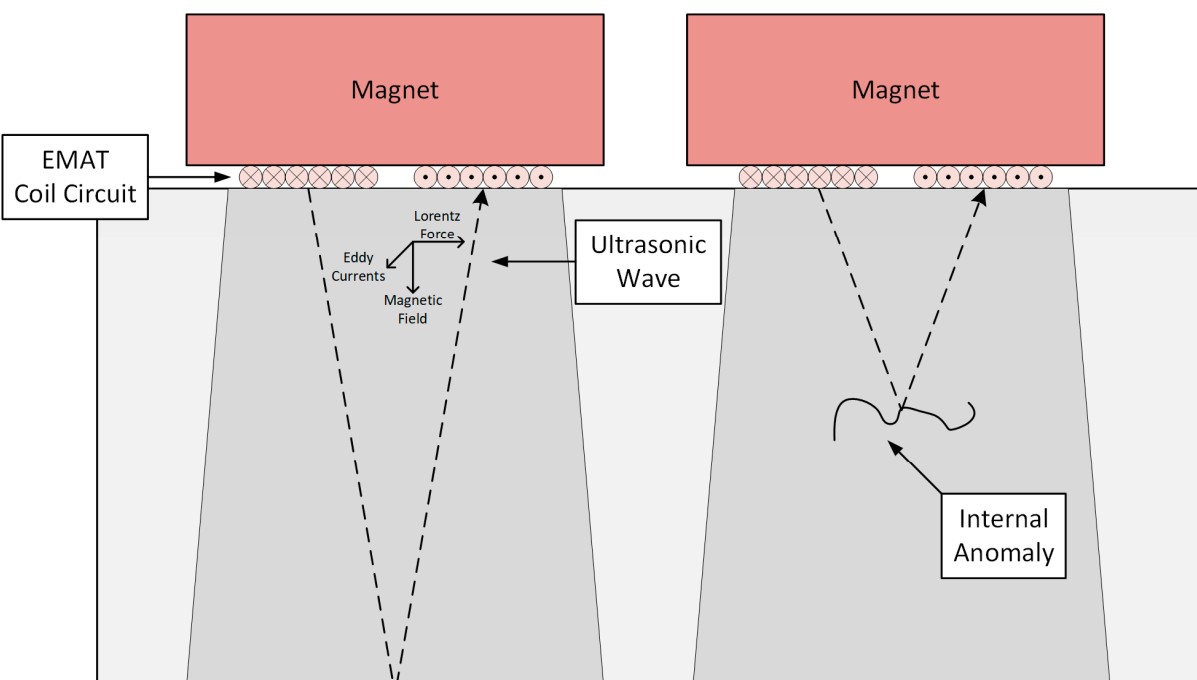

**Figure 4.** The working principle of an EMAT sensor.

EMAT technology is relatively new compared to other basic ILI technologies. EMAT does not require any coupling fluid. Hence, it can be used in both liquid and gas pipelines. Scanning is reliable since there is no requirement for coupling between the probe and the pipe wall, but the probe and the wall must be separated by a specific distance. The anomalies that EMAT can detect are blisters, laminations, cracking, wall thickness, etc. In addition, EMAT technology is known to be a reliable and accurate method for the detection, identification, and sizing of hybrid anomalies such as SCC [103].

EMAT and UT technologies have significant differences in methodology. While UT concentrates on classical wave modes generated by piezoelectric probes, EMAT is the most advanced technology concentrating on shear horizontal (SH) and guided waves [104].

Parameters such as amplitude and phase shift of these wave modes are important in the classification and sizing of anomalies. Hirao and Ogi [105] developed an EMAT technique to detect corrosion anomalies on the outer surfaces of steel pipelines and determined that the amplitude and phase shift of the $SH_1$ mode is more sensitive to the presence of anomalies than the $SH_0$ mode. In the study, round-trip signals of $SH_0$ and $SH_1$ modes have proven to be uniquely responsive to surface anomalies. Gauthier et al. [106] tested the probe's success on notches on a pipe, using multi-mode SH waves generated by EMAT. Zhao et al. [107] demonstrated that the $n_1$ mode SH wave generated by EMATs can successfully detect and classify mechanical dents at the outer surface of the pipe wall at a depth of 25% or more along the wall. Klann and Beuker [108] presented a study on the detection of cracks in steel pipelines using $SH_n$ waves produced by EMAT. The detection tool using $SH_n$ waves produced by EMAT was compared with the traditional MFL method. In anomaly classification, EMAT performed better than MFL. Cong et al. [109] proposed a new EMAT design based on a magnetostriction mechanism to generate and receive longitudinal guided waves. The proposed design has advantages such as small volume and light weight, which helps to increase inspection efficiency in anomaly detection in pipes.

Apart from wave modes, there are different parameters such as configuration that affect the anomaly-sizing success of EMAT. Tu et al. [110] introduced a different EMAT configuration called ring array to enlarge the detection range. Thanks to this configuration, the entire cross-section of the pipe can be scanned with a single solenoid coil and enough permanent magnets. The new configuration yielded successful results in detecting pipe-wall thickness and anomalies in the pipeline. In addition to configuration and wave mode, the lift-off distance of the EMAT is an important criterion. For the EMAT to work effectively, there must be a constant lift-off distance between the pipe's inner wall and the sensor. This distance causes the EMAT to be sensitive to noise [111]. Noise reduction is a critical step to increase the reliability of the EMAT system [112] and noise can be reduced using signal processing methods [113–118].

### 3.3. Eddy Current

The EC method is only applicable to conductive substances. When an EC sensor is used to test a gas pipeline or other conductive structure, the sensor generates an electromagnetic field that penetrates the surface of the material. When this magnetic field intersects with the conductive material, electromagnetic ECs are induced in the conductive material. If there are any defects or discontinuities in the material, such as cracks or corrosion, they will disrupt the flow of the ECs. The sensor can detect these disruptions and use them to identify the location and size of the anomalies. Figure 5 demonstrates the operation of an EC sensor.

The EC technique is frequently used for crack detection [119,120] due to the advantages arising from its methodology. EC also offers non-contact testing and no residual effects. However, as with EMAT technology, the lift-off effect is exhibited here due to the non-contact nature of the tool. The anomalies that EC can detect are cracks, laminar anomalies, etc.

Different EC imaging methods can be used to detect anomalies in metal structures. These methods are low-frequency EC imaging [121], multi-frequency EC imaging [122], transient EC imaging [123], and pulsed EC (PEC) imaging [124,125]. Among these imaging methods, PEC is frequently seen in the literature. By developing a feature-extraction algorithm based on principal component analysis, Tian et al. [126] demonstrated that more anomaly information is provided for PEC than traditional peak value and time. The new feature extraction algorithm eliminated the lift-off effect and proved effective in detecting cracks without scanning. Safizadeh and Hasanian [127] presented an advanced PEC technique for the detection of corrosion anomalies. The optimum test parameters were obtained by simulating the PEC test on a pipe with Maxwell software. The results obtained from the artificial anomalies produced on the inner surface of a gas pipe revealed that the lift-off effect was eliminated with the PEC technique and the corrosion was successfully

detected. Arjun et al. [128] presented a study on optimizing the PEC probe configuration for deeper penetration of the magnetic field in the material. The detection sensitivity of the optimized probe was investigated using notches machined at different depths, and the detection sensitivity of the PEC probe was increased. Tian et al. [129] proposed a new method for thickness measurement by analyzing the PEC detection system. The method does not need to evaluate electromagnetic parameters. This greatly simplifies the machine learning process, improves measurement accuracy, and can make the system more stable than traditional ways. Yu et al. [130] proposed an approach to reduce lift-off noise for detecting anomaly depth or width based on the investigation of the relationship between the peak value of the difference signal and lift-off. The proposed approach was validated by experiment and the results showed that lift-off noise can be greatly reduced in the PEC technique. Park et al. [131] developed PEC technology to detect the amount of thinning of a carbon steel pipe wall covered with insulation. The results showed that the PEC system could detect wall thinning in an insulated pipeline. Piao et al. [132] proposed a new high-speed PEC-detection method to detect internal/external anomalies using conductivity-dependent and permeability-dependent distribution models of EC induced in the inner surface of steel pipes. The method showed high inspection speed, good linearity, and superior sensitivity.

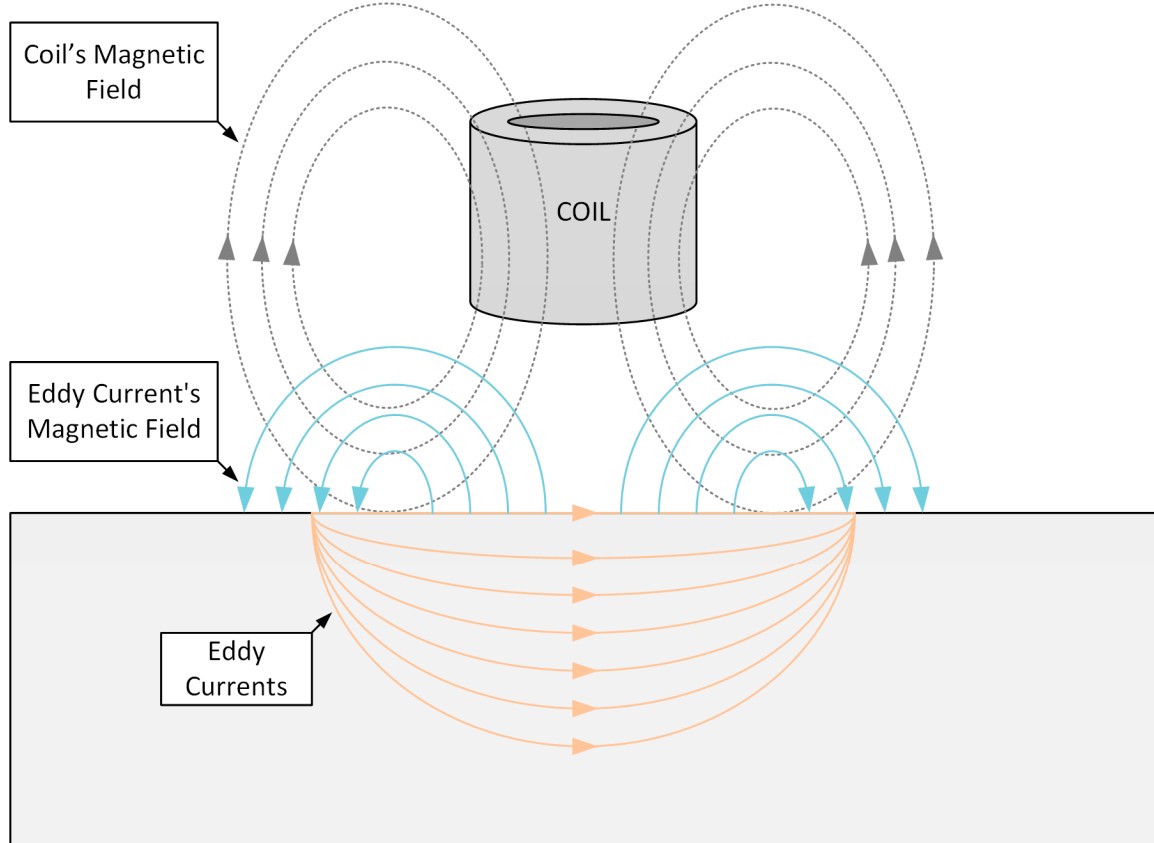

**Figure 5.** The working principle of an EC sensor.

EC can gain expertise in different types of anomalies over time because of its replaceable structure. For example, pipelines manufactured with corrosion-resistant alloy (CRA) cannot be inspected with UT or MFL. UT sensors cannot transmit the sound wave over the CRA liner. The MFL cannot inspect the pipe as it cannot penetrate the magnetic field due to the CRA liner. Asher and Boenisch [133,134] introduced a new ILI sensor technology based on the magnetic EC and multi-differential EC principles developed by Innospection Ltd. and ExxonMobil. This sensor was tested on anomalies artificially added to the CRA pipe, such as metal loss, erosion, internal girth weld, and crack-like defects. All anomalies were

detected with the sensors, including very small anomalies. Remote field EC (RFEC) has advantages such as being unaffected by the skin effect and material properties in anomaly detection of metal pipelines. However, the RFEC probe is large in size and the signal received by the sensing coil is weak. She et al. [135] proposed a new configuration for RFEC. Thanks to this new configuration, the probe size was reduced and the signal received by the sensing coil was strengthened.

### 3.4. Magnetic Flux Leakage

MFL sensors typically consist of a permanent magnet and a sensor coil, which are placed on opposite sides of the material being tested. The permanent magnet generates a magnetic field within the material. If there are any imperfections or anomalies in the material, they will disrupt the magnetic field and cause a leakage of magnetic flux. Leakage of magnetic flux is then detected by the sensor coil, allowing the presence and location of anomalies in the material to be determined. Figure 6 illustrates the operation of an MFL sensor.

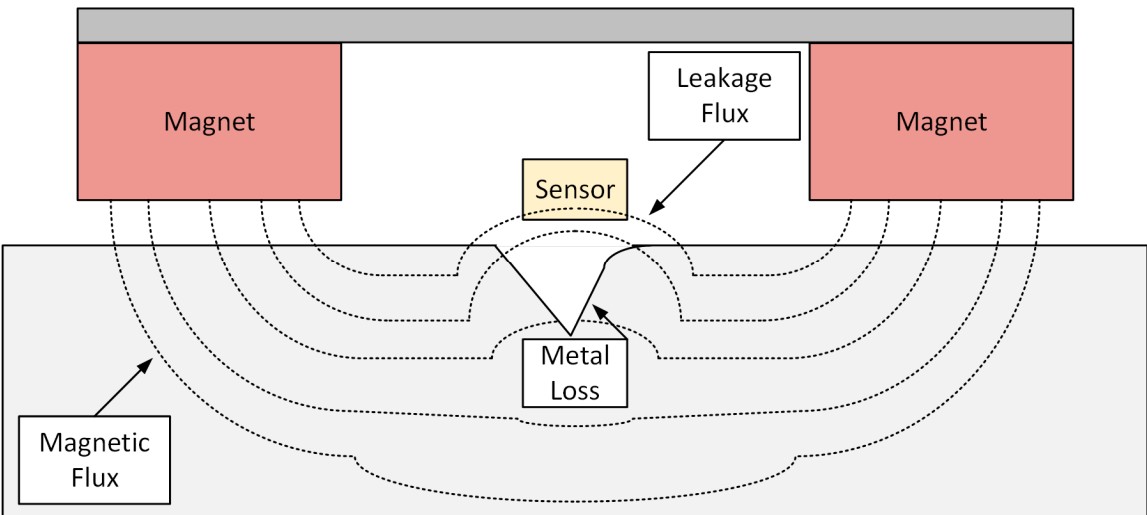

**Figure 6.** The working principle of an MFL sensor.

One of the oldest methods for detecting metal loss is the MFL method [136]. MFL is also the most common ILI technology. MFL sensors are a highly effective NDE tool for the inspection of ferromagnetic materials, such as steel pipes. MFL testing is relatively fast and inexpensive compared to other NDE methods, such as UT. It can conveniently determine the location and orientation of the anomaly and whether it is inside or outside the pipe. The anomalies that MFL can detect are metal losses, metallurgical changes, dents, etc.

The success of the MFL technique depends on many parameters. One of these parameters is a sensitive magnetic sensor. Pham et al. [137] developed the planar hall magnetoresistance magnetic sensor for use in MFL. The results showed improvements in the bipolar and linear responses to the magnetic field, high sensitivity, and low thermal drift. To find out which parameters the MFL signal is affected by, FEA is often used [138]. Chen et al. [139] studied MFL signals on four different corrosion anomalies with three-dimensional FEA. The findings showed that the relative position of the corrosion affects the amplitude of the MFL signal. In addition, the amplitude of the MFL signal is also affected by the morphology of the anomaly. Conventional MFL technology creates a magnetic field aligned to the axis of the pipe being inspected. Therefore, while the MFL can easily detect anomalies perpendicular to the field, it has difficulty detecting long and thin anomalies. In contrast to the traditional MFL structure, Kim et al. [140] presented a method called circumferential MFL (CMFL) or transverse field inspection (TFI) for the detection and characterization of signals produced by long and narrow cracks formed by the external–internal pressure difference. The findings showed that the circumferential magnetic fields maximized the leakage of

magnetic flux in cracks. Technology developed by T.D. Williamson Inc. [141] called spiral MFL (SMFL) possesses MFL and CMFL detection capabilities. The proposed MFL has become able to detect both perpendiculars to the field and long and thin anomalies. Okolo and Meydan [142] presented a quantitative approach based on the pulsed MFL (PMFL) method for the detection and characterization of signals produced by hairline cracks. The findings show that the proposed technique can be used to classify hairline cracks. One of the parameters affecting the success of the MFL technique is the processing of MFL inspection signals. Mao et al. [143] presented a detailed study on preprocessing and processing of MFL inspection signals. Carvalho et al. [144] used an artificial neural network (ANN) to classify signals collected along the weld bead as defective or non-defective. Again, ANN was used to classify signals labeled as defective as external corrosion, internal corrosion, and lack of penetration. The study showed a success rate of 94.2% for the first classification and 71.7% for the second classification. Ma and Liu [145] used the immune radial basis function neural network to process MFL signals. The location and size of the corrosion were successfully determined in the tests. Layouni et al. [146] presented a study to detect, locate, and estimate the size of metal-loss anomalies from MFL scans of oil and gas pipelines. Pattern-adapted wavelets were used for the anomaly length and ANN was used for the anomaly depth. The proposed technique is computationally efficient, provides a high level of accuracy, and works for a wide variety of anomaly shapes. In the processing of MFL signals, noise elimination is also a crucial step. Ji et al. [147] presented a noise-elimination algorithm called adaptive fuzzy lifting wavelet transform to solve the noise reduction problem in MFL signals. The findings show that this method achieves a better noise reduction than that obtained with conventional wavelet transform. Mukherjee et al. [148] suggested a new scheme of channel equalization algorithm to correct misalignments of MFL sensors, resulting in excellent signal recovery and noise elimination.

*3.5. Mechanical Contact*

MC tools contain a mechanical arm, a mechanical or magnetic encoder, and a spring system. The spring system pushes the mechanical arm against the pipeline's inner wall. The displacement of the mechanical arm in contact with any anomaly in the pipeline is read by the encoder as angular displacement and converted into depth information by processing. Figure 7 depicts the working principle of an MC sensor.

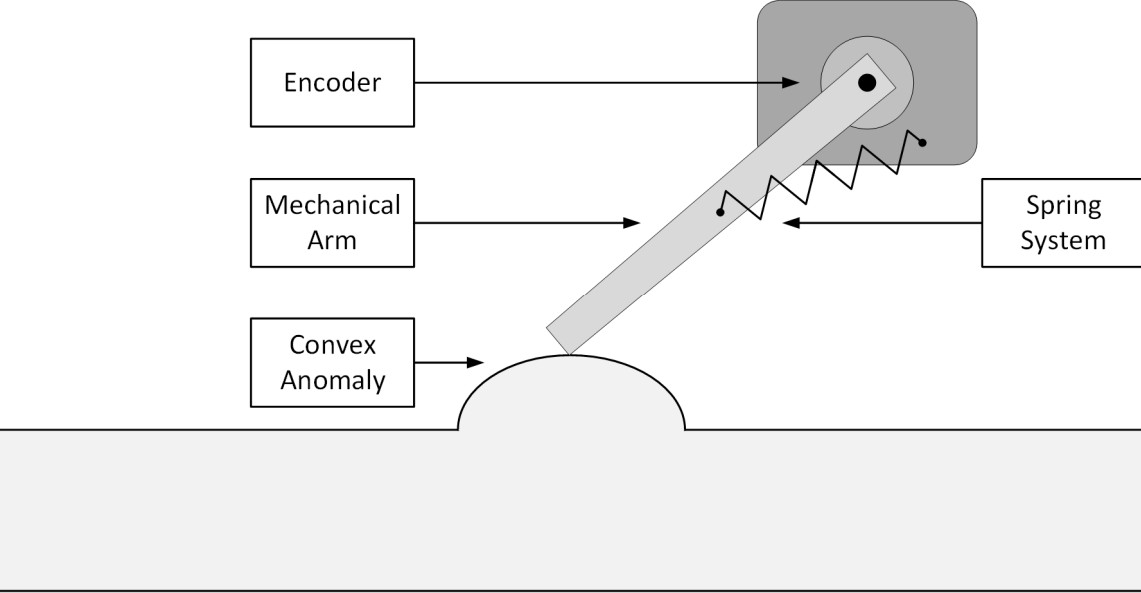

**Figure 7.** The working principle of an MC sensor.

The majority of MC tools are used to identify pipeline geometry. Besides their ability to measure the diameter and roundness of a pipeline, MC sensors can detect any geometric anomalies in the pipe wall. MC sensors can also measure the distance between the pipeline wall and any objects that may be present within the pipe, such as deposits or debris. This can help identify potential blockages or obstructions that may cause problems in the pipeline. The anomalies that MC can detect are dents, welds, buckles, wrinkles, ovality, and pits, among other irregularities.

In MC tools, choosing an encoder is crucial. Mechanical encoders can have several issues in terms of sensitivity to collision and vibration and excessive power consumption. For this reason, magnetic encoders are generally preferred in MC tools. Kim et al. [149] developed a high-resolution, low-power-consumption, fast-response MC tool using a non-contact rotary sensor produced using an annular anisotropic magnet and a hall effect sensor. Later, Kim et al. [150] designed a 30″ geometry pig using this MC tool. The pig included the inertial measurement unit (IMU), odometer, and MC tool. The MC tool was used to measure ovality, dent size, and pipeline inside diameter, the IMU for trajectory measurement and three-dimensional coordination in space, and the odometer for distance traveled and instantaneous velocity measurement. Xiaolong et al. [151] gave general information about the design principles of the MC tool (mechanical structure, mathematical model, sensor circuit design, methodology, etc.). In addition, the basic components of the sensing arm were optimized by analyzing the magnetic field strength and structural strength. Canavese et al. [152] designed a new low-cost and low-risk foam-made pig that can detect, position, and size internal diameter changes and anomalies. In this design, a strain sensor is used to detect the anomaly depth. Laboratory results showed successful sizing of internal diameter changes and corrosion. The results obtained in the field test [153] were compared with the results provided by the commercial conventional MC tool launched into the same pipeline under the same conditions. The developed pig provided more information about the pipeline structure than the commercial one.

One of the disadvantages of the traditional MC tool is the dynamic behavior of the mechanical arms under operating conditions. There are three different types of mechanical arms [154], called a wheel, arm, or probe, and they all have different dynamic behavior. Li et al. [155] tested the dynamic behavior of the probe-type mechanical arm. Analytical and experimental results show that pigging speed and spring force are closely related to mechanical arm sensitivity. Zhu et al. [156] tested the dynamic behavior of the wheel-type mechanical arm. According to the experimental results, high pigging speed increases the measurement error, while low pigging speed reduces the measurement error. Increasing the spring force can reduce or even eliminate the measurement error. Li et al. [157] proposed a bouncing model that can be used to calculate the bouncing height and sliding length of the wheel-type mechanical arm along the convex defect. According to the model, it was observed that the bouncing was caused by the sudden angular acceleration, and the sudden angular acceleration was caused by the anomaly shape and rigid collision. Paeper et al. [158] designed a non-contact measurement technology as a solution to the dynamic disadvantage of the traditional mechanical arm. This technology was a mechatronic arm with a conventional mechanical arm and a touchless operating proximity sensor. As a result, geometric anomalies in the pipe, such as dents or wrinkles, are detected by the sensor, which offers comprehensive shape information.

## 4. Anomaly-Detection Capabilities of In-Line Inspection Sensors

ILI sensors are used to detect and size anomalies such as metal loss, cracking, and geometric deformation. The basic ILI sensors cannot exhibit the same resolution and accuracy for every anomaly type, or even detect some anomalies. This is known as the anomaly-detection capability of that sensor. In this section, the anomaly-detection capabilities of UT, EMAT, EC, MFL, and MC sensors are given, and basic sensors used in the market and some hybrid models are compared in terms of resolution and accuracy.

Table 1 summarizes the basic ILI sensors' capability for detecting anomalies. These sensors are not capable of providing the same level of resolution and accuracy for all types of anomalies. Comparing the two commercially available UT [159] and EMAT [160] tools, the minimum length that the UT tool can detect is 25 mm, while the minimum length that the EMAT tool can detect is 40 mm in crack detection. Although the minimum depth it can detect in a long seam in both tools is 2 mm, the length sizing of the UT tool is ±10 mm and the length sizing of the EMAT tool is ±20 mm. The findings show that the UT tool performs better at detecting and sizing cracks. In metal-loss detection, the UT [161] tool's length sizing accuracy is ±7 mm, the MFL [162] tool's length-sizing accuracy is ±15 mm, and the EC [163] tool's length-sizing accuracy is ±6 mm. Width-sizing accuracies are ±8 mm for UT tool, ±15 mm for MFL tool, and ±5 mm for EC tool. The findings show that the EC tool performs best at detecting and sizing metal losses. However, MFL tools are generally preferred for detecting and sizing metal loss, as the UT tool is limited to liquid mediums and the EC tool is unable to measure the metal loss on the pipe's outer surface or wall thickness. ILI technologies used in the inspection of natural gas and oil pipelines are sometimes used as hybrids to combine their advantages or eliminate their disadvantages. In metal-loss detection, the MFL-UT [164] hybrid tool's depth-, length-, and width-sizing accuracy is ±0.4 mm, ±7 mm, and ±8 mm, respectively, while the MFL-EC [165] tool's is ±1.3 mm, ±6 mm, and ±5 mm, respectively. The findings showed that the MFL-UT hybrid tool was better at depth sizing and the MFL-EC hybrid tool was better at length and width sizing.

**Table 1.** Detection capabilities of the basic ILI sensors.

| Tool Type | Metal Loss | | Cracking | | Geometric Deformation | |
|---|---|---|---|---|---|---|
| | Internal Corrosion | External Corrosion | Axial Crack | Circumferential Crack | Dent | Ovality |
| UT | * [166–168] | * [167–169] | * [170–172] | * [173–175] | * [176–178] | |
| EMAT | * [179–181] | * [182,183] | * [184–186] | * [187–189] | * [107,190] | |
| EC | * [191–193] | | * [194–196] | * [197,198] | * [199–201] | |
| MFL | * [202,203] | * [204–206] | * [207,208] | | * [209,210] | |
| MC | | | | | * [211–213] | * [214–216] |

* indicates capability.

Apart from the measurement sensitivity of the smart pig, sometimes hybrid pigs are preferred for anomaly classification. For instance, while the MFL sensor can detect metal losses on both the inner and the outer surfaces of the pipe, it cannot determine whether the damage is on the inner or outer surface. In this case, due to the presence of hybrid structures such as (MFL + EC), it is possible to deduce that the EC-detected metal losses are internal. Dual MFLs can also be used for the same purpose. While the first MFL saturates the entire pipe, the second MFL with a lower magnetic field strength can only saturate half the pipe. Therefore, anomalies that can be detected by both MFLs are labeled as internal. In addition, MC tools are inefficient for detecting metal loss and cracks, but they are unique in detecting dents and ovality as they are in direct contact with the pipe's inner wall. Thus, MC tools are frequently used in hybrid pigs.

## 5. Impact of In-Line Inspection Sensors on Environmental Issues

Anomalies such as geometric deformation, metal loss, and cracking occur in pipelines for various reasons. These anomalies cause rupture of pipelines, hence leakage. Sometimes anomalies trigger other anomalies to occur. Hafez [217] studied the case of an oil spill that occurred in an undersea crude oil pipeline. According to the case outputs, a plaint dent formed in the pipeline, presumably by external effects, caused corrosion and cracking, respectively, and caused the pipeline to rupture. Pipeline leaks have occurred in Russia [218], Peru [219], Canada [220], and the USA [221] in recent years. These leaks caused thousands of barrels of oil to leak and pollute the environment.

Pipeline accidents have considerably decreased as a result of the development of ILI technology. In fact, most pipeline accidents in the last decade have not occurred on ILI-applied liquid or gas transmission pipelines, but on non-ILI-applied local gas distribution systems [222]. This has resulted in a major reduction in the volume of hazardous liquid leaks.

Siler-Evans et al. [223] analyzed data on pipeline incidents that took place in the USA from 1968 to 2009 and found that the number of hazardous liquid pipeline accidents has dropped significantly over the past four decades, resulting in a fourfold reduction in the annual volume of hazardous liquid leaks. PHMSA's statistics on oil pipeline accidents affecting people or the environment in the USA between 2010 and 2020 also support this. The volume spilled rate per billion barrel-miles transported is given in Figure 8. Although the volume spilled rate per billion barrel-miles transported has fluctuated since 2010, the trend line is in a downward trend.

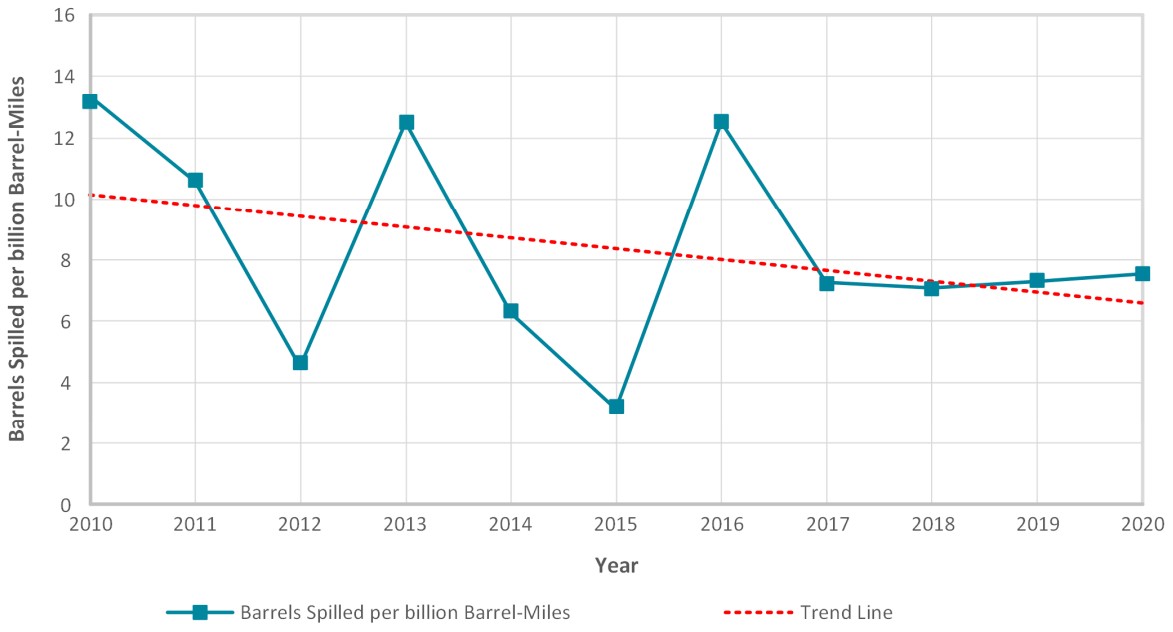

**Figure 8.** Barrels spilled per billion barrel-miles between 2010 and 2020.

## 6. Conclusions

ILI is crucial to the sustainability of the oil and gas pipeline industry. Advances in ILI technology will lead to more accurate detection of anomalies in the pipeline and better prediction of the growth rate of anomalies and the life of the pipeline, preventing pipeline ruptures and eliminating the potential vital, environmental, and financial damages of rupture. This paper discusses the potential of ILI technology used in natural gas and oil pipelines. The most common types of anomalies in oil and natural gas pipelines are mentioned, and the causes of anomalies and anomaly management are emphasized. By mentioning the basic ILI sensors, the methodologies, historical developments, limitations, and studies of the sensors in the literature are presented. The sensors were compared according to their anomaly-detection capabilities, and the accuracy and resolution information of the basic and hybrid sensors used in the sector were given. The effect of the use of ILI tools on the integrity of the pipeline was examined, and the trends in the number of accidents and the amount of harmful liquid leaking into nature were examined.

The growing demand for energy has resulted in the construction of new pipelines all over the world. According to PHMSA's data, a total of 107,835 miles of gas pipelines were constructed in the USA between 2010 and 2021. Although significant lengths of pipelines have been constructed worldwide, the rate of serious accidents has been reported to be relatively low [224]. It is undeniable that ILI tools account for a significant portion of this

decline. For this very reason, ILI tools seem to have a bright future in PIM, such that the smart pig market, which was worth USD544.7 million in 2017, is expected to be worth USD717.9 million in 2023 and USD900 million in 2027 [225]. As worldwide industrialization continues, the world's energy needs will increase. With the need for energy, the extension of pipelines will become inevitable. Because increasing the length of pipelines increases the chances of vital, environmental, or financial consequence of pipeline rupture, ILI technology must continue to advance.

The most in-demand ILI sensor technologies in the industry are still the MFL for metal losses, the UT for cracks, and the MC for geometric deformations. However, due to the advantages it brings, EMAT technology is becoming more popular among ILI methods and is likely to dominate the smart-pig market in the future. EMAT has the potential to provide more detailed information about pipeline conditions than other inspection methods and does not require a couplant, as it uses electromagnetic waves to generate the ultrasound signal. Additionally, EMAT is effective at detecting metal losses and is capable of detecting other types of anomalies, such as cracks and geometric deformations. The non-contact nature of EMAT inspections allows for inspections to be performed without the need for pipeline cleaning or shutdown, saving time and resources. Although EMAT saves time and resources, in general, smart pigs are inherently problematic in terms of time and resources due to pipeline preparation, the complexity of data analysis, and becoming stuck in the pipeline. For this reason, the ILI market is likely to be shared by new and inexpensive products such as inspection balls in the future.

**Author Contributions:** Conceptualization, H.A.Y.; methodology, H.A.Y.; evaluating literature, B.O.P.; validation, B.O.P.; formal analysis, B.O.P.; visualization, B.O.P.; resources, B.O.P.; writing—original draft preparation, B.O.P.; writing—review and editing, H.A.Y.; supervision, H.A.Y. All authors have read and agreed to the published version of the manuscript.

**Funding:** This research received no external funding.

**Institutional Review Board Statement:** Not applicable.

**Informed Consent Statement:** Not applicable.

**Data Availability Statement:** Not applicable.

**Conflicts of Interest:** The authors declare no conflict of interest.

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
