# Peer review of "A Comprehensive Analysis of In-Line Inspection Tools and Technologies for Steel Oil and Gas Pipelines"

_sustainability, doi:10.3390/su15032783_

Round 1

Reviewer 1 Report

The manuscript is a literature review on in-line inspection technologies for oil and gas pipelines.  I do not work on pipelines and I will just give my comments from the view of an non-expert reader. I have noted the following issues.

1. Acronyms of technologies UT, EMAT, MFL and EC are introduced in the Abstract. Then these acronyms are used in Section 2 before they are described in Section 3. Introducing acronyms in abstract cannot replace description of them before using them.

2. There is a lack of details. Many references are given without description or critical review of the work. For example, in the first paragraph of page 3 it is stated that pipelines are calssified as piggable or unpiggable. Pig operations can be conducted without difficulty in pipelines that are piggalbe and “piggable lines may become un-piggable  over time for various reasons [33]”. For the readers’ sake, it would be better to state what these various reasons are, or at least what they include. As another example, in the second paragraph of Section 3 it is stated that “Sampath et al. [77] introduced a new pig-adapted non-contact optical sensor array method for real-time inspection and non-destructive evaluation (NDE) of pipelines. The new array was successful on deposits, cavities, and uniform corrosions”. It is not clear what this method is, unless the readers find and read the referenced article.

3. In the Introduction it is stated that “Section 2 provides the causes, hazard potentials, and management methods of pipeline anomalies”. However, the management methods are not described in Section 2. What are these methods? If anomalies are detected, what will be done? Will replacement of anormal parts/sections be necessary or will repairment be possible?

I hope the authors will consider the above issues.

Reviewer 2 Report

Thanks much for the authors’ work in making a nice review of in-line inspection techniques. The paper is well outlined and easy to read, and it covers almost all for a good review paper. I’d like to give some comments for the authors to address and revise before publication.

  1. I suggest to have a Table of in-line sensing techniques, comparing them on items like principles, sensing device (dimension, etc), types of anomalies applications, pros and cons, impact, etc, which are partly described in Session 4 and 5. (Table I is good, but didn’t give direct information, and not as comprehensive as the one I suggested, but please keep your Table I too)
  2. Figure 8: blue curve, I suggest to use symbol+line.
  3. In the conclusion, or after that, please have an outlook of the in-line inspection techniques, like which will be more widespread, and what kind of new technologies can be applied in the inspection, et al.
